# The Mystery of Red Blood Cells Extracellular Vesicles in Sleep Apnea with Metabolic Dysfunction

**DOI:** 10.3390/ijms22094301

**Published:** 2021-04-21

**Authors:** Abdelnaby Khalyfa, David Sanz-Rubio

**Affiliations:** 1Department of Child Health and the Child Health Research Institute, University of Missouri School of Medicine, Columbia, MO 65201, USA; 2Translational Research Unit, Hospital Universitario Miguel Servet, Instituto de Investigación Sanitaria de Aragón (IISAragón), 50009 Zaragoza, Spain; davidsanzrubio91@gmail.com

**Keywords:** red blood cells, RBCs, extracellular vesicles (EVs), exosomes, metabolic dysfunction, sleep disordered, therapy, clinical application

## Abstract

Sleep is very important for overall health and quality of life, while sleep disorder has been associated with several human diseases, namely cardiovascular, metabolic, cognitive, and cancer-related alterations. Obstructive sleep apnea (OSA) is the most common respiratory sleep-disordered breathing, which is caused by the recurrent collapse of the upper airway during sleep. OSA has emerged as a major public health problem and increasing evidence suggests that untreated OSA can lead to the development of various diseases including neurodegenerative diseases. In addition, OSA may lead to decreased blood oxygenation and fragmentation of the sleep cycle. The formation of free radicals or reactive oxygen species (ROS) can emerge and react with nitric oxide (NO) to produce peroxynitrite, thereby diminishing the bioavailability of NO. Hypoxia, the hallmark of OSA, refers to a decline of tissue oxygen saturation and affects several types of cells, playing cell-to-cell communication a vital role in the outcome of this interplay. Red blood cells (RBCs) are considered transporters of oxygen and nutrients to the tissues, and these RBCs are important interorgan communication systems with additional functions, including participation in the control of systemic NO metabolism, redox regulation, blood rheology, and viscosity. RBCs have been shown to induce endothelial dysfunction and increase cardiac injury. The mechanistic links between changes of RBC functional properties and cardiovascular are largely unknown. Extracellular vesicles (EVs) are secreted by most cell types and released in biological fluids both under physiological and pathological conditions. EVs are involved in intercellular communication by transferring complex cargoes including proteins, lipids, and nucleic acids from donor cells to recipient cells. Advancing our knowledge about mechanisms of RBC-EVs formation and their pathophysiological relevance may help to shed light on circulating EVs and to translate their application to clinical practice. We will focus on the potential use of RBC-EVs as valuable diagnostic and prognostic biomarkers and state-specific cargoes, and possibilities as therapeutic vehicles for drug and gene delivery. The use of RBC-EVs as a precision medicine for the diagnosis and treatment of the patient with sleep disorder will improve the prognosis and the quality of life in patients with cardiovascular disease (CVD).

## 1. Sleep Apnea and Metabolic Dysfunction

Sleep is an integral part of life, with humans spending approximately one third of our lives asleep. Poor sleep quality and reduced amounts of sleep can clearly result in daytime sleepiness, decreased alertness, and reduced mental functioning, mainly psychiatric disorders such as stress or depression [1]. Sleep-disordered breathing (SDB) is affecting both genders and frequently associated with a wide variety of co-morbid disorders in multiple organs systems [2]. The major categories included in SDB consist of obstructive sleep apnea (OSA), central sleep apnea (CSA), obesity hypoventilation syndrome (OHS), and sleep-related hypoxemia such as in COPD and other lungs parenchymal disease [3]. Furthermore, OSA is the most common form of SDB and is associated with many adverse health consequences with increased overall mortality [4,5,6]. OSA is characterized by partial or complete recurrent upper airway obstruction during sleep, resulting in periods of hypopneas and/or apneas, oxyhemoglobin desaturation, and frequent night awakenings with excessive daytime sleepiness, and consequently, reduced performance in social activities. OSA is associated with an increased risk of end-organ morbidities and overall mortality and imposes adverse health consequences. Each episode of apnea, hypopnea, or both is usually followed by a marked decrease in arterial oxygen saturation. It has been reported that OSA increases with age and is approximately twice as common in men as in women. The prevalence of OSA is approximately 26.6% in men and 8.7% in women among individuals aged from 30 to 49 years and approximately 43.2% in men and 27.8% in women among individuals aged from 50 to 70 years [7].

The cardiovascular morbidities resulting from OSA have also been studied broadly in animal models to further elucidate the underlying molecular mechanisms. OSA had an increased risk for the development of myocardial infarction, coronary revascularization events, and cardiovascular death, independent of other cardiovascular risk factors [8], and it has been implicated as an independent risk factor for the development of coronary artery, hypertension, stroke, disease, cardiac arrhythmia, congestive heart failure, pulmonary hypertension, and aortic dissection [9,10,11]. The prevalence of cardiovascular diseases (CVDs) is projected to rise to about 41% of the USA population by 2030 [12]. Furthermore, intermittent hypoxia (IH) associated with OSA can lead to atherosclerosis by creating endothelial dysfunction, endothelial injury, oxidative stress, vascular inflammation, hyperlipidemia, hypercoagulation, and sympathetic activation [13]. IH can activate several global signaling pathways, which play a key role in mediating the inflammatory and cardiovascular consequences in OSA. Continuous positive airway pressure (CPAP) is considered the gold standard treatment with OSA patient, and this treatment is marginally effective in reducing elevated arterial pressure (~2 mmHg), and sympathetic nerve activity and partially restoring chemoreflex and baroreflex sensitivity [14,15]. However, the use of CPAP was not associated with reduced risks of cardiovascular outcomes, diabetes mellitus, or death for patients with OSA in recent randomized controlled trials [16,17,18]. Therefore, it is essential to develop novel pharmacological agents to counteract the pathophysiological mechanisms responsible for OSA-related adverse consequences, namely oxidative stress, sympathetic activation, and low-grade inflammation [16].

It has been indicated that IH increases the number of multipotent hematopoietic progenitors, promotes erythropoiesis, and increases monocyte counts [19]. The long term of IH can lead to a decrease in cardiac systolic function, and patients with severe chronic obstructive pulmonary disease may have changes in hemorheology, endothelial function, coagulation, and the fibrinolytic system [20,21,22]. Patients with sleep apnea and cardiovascular disease share many risk factors such as hypertension, dyslipidemia, diabetes mellitus, and obesity. IH during sleep is associated with a number of pathological responses including oxidative stress, inflammation, and sympathetic activation [23]. The physiological and pathological responses to these different types of hypoxia are different, but the underlying mechanisms are not fully understood.

Cardiovascular diseases, such as coronary artery diseases, heart failure, and stroke, have a high prevalence and annually increasing incidence with high mortality and morbidity. Although treatment for CVDs have made great progress in the past decades, the few years survival rate for CVDs patients fails to be considerably improved. Through the increasing heterogeneity and complexity perceived in the progression of CVDs, the need for specific and accurate diagnosis of disease state and molecular monitoring of disease progression has become more urgent. Besides that, identifying a biomarker with high sensitivity and specificity for assessing the prognosis of CVDs is also necessary for optimizing personalized treatment and reducing mortality. Therefore, intense CVDs research continues to grow with the hunt for developing more targeted treatment options that promote the avenues of personalized medicine.

## 2. Mechanisms of Metabolic Dysfunction

Metabolic syndrome is a cluster of disorders that occur together, increasing your risk of heart disease, stroke, and type 2 diabetes, and these conditions include increased blood pressure, high blood glucose, excess body fat around the waist, and abnormal cholesterol or triglyceride levels [24]. About 20% of the adult population in Western countries are affected by metabolic syndrome [25]. Endothelial dysfunction is driven primarily by nitric oxide (NO) deficiency and increased bioavailability of oxidizing reactive oxygen species (ROS) [26]. Increased oxidative stress by ROS can impair NO bioavailability in plasma of coronary artery disease (CAD) patients, and NO synthase expression and activity were reduced in CAD red blood cells [27]. NO is a signaling molecule with a crucial role in regulating vascular tone. Endothelial dysfunction, with reduced NO bioavailability, is a pathological condition frequently occurring in CAD patients [28]. ROS are associated with a high number of different pathologies because they cause damage to biomolecules (lipids, proteins, and DNA) and subcellular structures; however, ROS are also involved in maintaining physiological conditions [29,30]. NO enhances vasodilatation, reduces platelet aggression and adhesion (anti-thrombotic), prevents smooth muscle proliferation, inhibits adhesion of leukocytes and expression of pro-inflammatory cytokines genes (anti-inflammatory), and counters the oxidation of low-density lipoprotein (LDL) cholesterol [26]. The enzymatic production of NO by endothelial NO synthase (eNOS) is crucial in mediating endothelial function, and oxidative stress can cause dysregulation of eNOS and endothelial dysfunction. Studies in animal models and cultured cells have identified a key role of oxidative stress during the development of T2D vascular complications [31,32]. NADPH oxidases (NOX) are membrane-bound enzyme complexes that are considered potent stimulators of ROS in endothelial cells, particularly under conditions of hyperglycemia [33]. Reduced NO bioavailability might lead to inhibition of vasodilatory response and blood flow, insufficient tissue oxygenation in the microcirculation, and end-organ injury, especially in recipients with endothelial dysfunction issues [34,35]. Deficiency of NO has been associated with endothelial dysfunction and adverse cardiovascular outcomes [36]. The mechanisms linking OSA and CVDs are shown in Figure 1. Intermittent hypoxia caused by OSA alters insulin sensitivity, blood pressure, blood viscosity, increased systemic inflammation, and ROS, which lead to endothelial dysfunction and cardiovascular diseases. Furthermore, another proposed molecular mechanism is hypoxia-inducible factor 1 (HIF-1α) which is based on the oxygen-sensitive α-subunit of HIF-1α -a key regulator of oxygen metabolism that plays an important role in OSA and metabolic dysfunction [37]. It has been indicated that HIF-1 is highly involved in mediating development and progression of type 2 diabetes (T2DM) though hypoxia [37]. In addition, HIF-1 is involved in the effects of IH on blood cells [38]. Several studies in OSA animal and cell models have been shown an increased expression of HIF-1a [39,40]. Other investigation showed that OSA patient increased HIF-1a levels in comparison to healthy control [41,42,43,44,45].

## 3. Red Blood Cells

Red blood cells (RBCs) are required during all stages of life—embryonic, fetal, neonatal, adolescent, and adult. In adults, the human body contains about 5 L of blood, and once blood is pumped out of the heart, it takes from 20 to 30 s to make a complete trip through the circulation and return to the heart [46]. Blood cells are formed through a process called hematopoiesis, and there are three types of blood cells such as RBCs, white blood cells, and platelets. RBCs (erythrocytes) carry oxygen to all cells in the body, while white blood cells (leukocytes) are an important part of the immune system, and platelets (thrombocytes) make the blood clot and help stop bleeding. RBCs are much more common than the other blood particles, of which there are about 4000–11,000 white blood cells and about 150,000–400,000 platelets per microliter [47]. For example, human RBCs take on average 60 s to complete one cycle of circulation, and approximately 2.4 million new erythrocytes are produced per second in human adults [48]. Blood cells are made in the bone marrow, but there are other organs and systems that help regulate blood cells including the lymph nodes, spleen, and liver [49]. Tissue oxygen delivery is a function of blood oxygen content and regional blood flow, and the volume and distribution of regional blood flow is actively regulated to maintain dynamic coupling between oxygen delivery and tissue respiration [50].

RBCs can act as both sensors and transducers by regulating the bioavailability of vasoactive effectors in plasma [51]. These RBCs were considered for a long time as “bags” packed with hemoglobin circulating in the blood for the only purposes of gas exchange and maintenance of acid/base equilibria; however, the evidence is now accumulating that RBCs’ function is far more complex and highly regulated [52]. Furthermore, RBCs also deliver carbon dioxide to the lungs. These RBCs have a unique structure; for example, they contain large amounts of hemoglobin protein, which is responsible for the characteristic red color of blood. Healthy RBCs are round, and they move through small blood vessels to carry oxygen to all parts of the body. Abnormal hemoglobin, called hemoglobin S, causes sickle cell disease (SCD), and about 1 in 13 black or African American babies is born with sickle cell trait [53]. Erythrocytes are involved in the regulation of the cardiovascular system via mechanisms that include their interaction with the endothelium, and these mechanisms which include the export of NO-like bioactivity, and ATP exert important cardiovascular effects [54,55]. The gas transportation capacity of RBCs is not only determined by the characteristics of hemoglobin but also by the capacity to regulate intracellular pH, ATP production, redox status, resistance to osmotic and mechanical stress, and recognition and removal by the immune system [56].

NO plays several roles, including as a signaling molecule which has a pivotal role in regulating vascular tone and dilates all types of blood vessels by stimulating soluble guanylyl cyclase, increasing cGMP in smooth muscle cells [57]. RBCs are the most abundant blood cell population carrying eNOS isoform and represent the major storage compartment of circulating NO metabolites [58]. Several studies have analyzed the role of RBCs in different cardiovascular diseases where the presence of reduced NO levels is one of the main pathological features [27,58]. One potential mechanism contributing to the reduced NO bioavailability is increased oxidative stress, due to an enhanced production of ROS coupled with decreased degradation of ROS through the presence of dysfunctional enzymatic antioxidant defense mechanisms. In addition, augmented ROS (such as superoxide) production reduces NO bioavailability by readily reacting with newly synthesized NO, forming peroxynitrite, which further potentiates superoxide production by uncoupling eNOS [57]. The mechanisms of interaction between RBCs and the endothelium are unknown.

Human RBCs express arginase, and endothelial cell arginase has emerged as an important regulator of NO production by competing with eNOS for their common substrate L-arginine [59,60]. RBC deformability was found to be decreased in several disease states associated with oxidative stress and endothelial dysfunction and/or impaired NO bioavailability, such as hypertension and diabetes [61,62]. It has been reported that RBCs injury due to energy or antioxidant depletion causes the breakdown of membrane phosphatidylserine asymmetry, with consequent exposure of phosphatidylserine at the erythrocyte surface (eryptosis) and binding to the phosphatidylserine receptors at macrophages and liver Kupffer cells [63].

Over the last decades, RBCs transfusions have been used in clinical practice and represent the most common therapeutic procedure performed in human patients in many diseases [64]. Depending on the storage solution and national regulatory requirements, RBCs can be stored in a refrigerator for up to 42 days before transfusion [65,66]. RBCs during refrigeration can undergo several multiple metabolic and structural changes [67]. Studies utilizing healthy murine models and human subjects have reported similar results concluding that increasing hemolysis of transfused RBCs in proportion to increasing storage time before transfusion [68,69]. RBCs contribute to vascular function and integrity, in addition to their function as oxygen transporters [70].

## 4. Red Blood Cells and Cardiovascular Diseases

Cardiovascular disease is the major leading pathological cause of morbidity and mortality worldwide [71,72,73]. Several epidemiological studies indicated a possible association between hemorheological profile and CVDs [74]. Furthermore, cardiovascular abnormalities are common in sickle cell anemia (SCA), including cardiac enlargement, myocardial infarction, acute stroke, chronic cerebral ischemia, arrhythmias, increased arterial stiffness, and microcirculation damage due to vaso-occlusive crisis, QT interval borderline or moderate prolongation, and cardiac autonomic neuropathy [75,76]. Increased white blood cell count together with elevated plasma fibrinogen levels, and hematocrit increases the resistance to blood flow [77].

Anemia is a known risk factor for ischemic heart disease, and reduced oxygen transport capacity in anemia causes a compensatory increase of the heart rate, resulting in a shorter myocardial perfusion time in diastole [78,79]. Furthermore, anemia increases morbidity and mortality in CVDs due to compensatory consequences of hypoxia, such as a hyperdynamic state with increased cardiac output, left ventricular hypertrophy, and progressive cardiac enlargement, and, probably, a proatherogenic role. In addition, anemia causes hypoxia due to decreased hemoglobin levels, and there are several nonhemodynamic and hemodynamic compensatory mechanisms [80]. It has been suggested that arterial dilatation involves to the recruitment of new vessels and formation of collaterals and arteriovenous shunts, hypoxic vasodilation due to hypoxia-generated metabolites, flow-mediated vasodilatation, and endothelium-derived relaxing factor [81,82,83]. Anemia increases cardiac output, may lead to eccentric left ventricular hypertrophy, activation of the sympathetic nervous system, and stimulation of the renin angiotensin aldosterone system, and is closely associated with chronic inflammation and increased oxidative stress [83,84].

SCA is the most common and most severe form of SCD, a group of inherited blood disorders caused by a genetic mutation due to vaso-occlusion of small blood vessels. The primary cause of the clinical phenotype of SCA is the intracellular polymerization of sickle hemoglobin resulting in sickling of RBCs in deoxygenated conditions [85]. Patients with SCA produce more ROS than healthy individuals that causes further damage to RBCs membrane [86]. Another clinical phenomenon associated with SCA is the dehydration of RBCs resulting in the formation of dense cells [87]. The alterations in cell geometry and membrane properties of RBCs could lead to impaired functionality including loss of deformability [88]. For instance, the shape distortion and membrane stiffening of RBCs induced by parasitic infectious diseases such as malaria and certain genetic blood disorders such as SCD cause increased cell rigidity and decreased cell deformability [89,90].

Several possible mechanisms for the role of RBCs in coronary heart disease have been suggested. Those include viscosity, increased platelet aggregation associated with the release of adenosine diphosphate, association with elevated serum cholesterol and triglycerides, deposition of cholesterol in the atherosclerotic plaque, stimulation of an excessive influx of macrophages, enlargement of the atherosclerotic necrotic core, and decreased fluidity of RBCs [91]. RBCs play a critical role in cardiovascular homeostasis by contributing to vascular function and integrity independently from their function as oxygen transporters [70]. RBCs are known to be involved in the physiological regulation of vascular tone via mechanisms involving release of ATP and generation of NO bioactivity [92], and NO in RBCs may be formed from the reduction of nitrite, primarily by deoxyhaemoglobin, or produced from eNOS [93,94]. RBC-derived NO bioactivity is involved in the generation of vascular relaxation under hypoxic conditions, which were observed in endothelial cells

## 5. Red Blood Cells Extracellular Vesicle

Mature RBCs are generated from multipotent hematopoietic stem cells to produce a highly functional specialized cells, and erythropoiesis occurs mostly in bone marrow and transfer into blood stream [95]. Here, we show erythroblasts undergo proliferation and terminal differentiation into non-nucleated reticulocytes then released from the bone marrow into the blood circulation where they mature into erythrocytes which contain microvesicles and exosomes Figure 2. RBC-EVs are formed only during the development of RBCs in bone marrow and are released following the fusion of multivesicular bodies (MVB) with the plasma membrane [96,97]. Morphological changes to RBCs in stored packed-RBC units were accompanied by shedding and release of EVs from RBCs or from residual platelets and leukocytes in the bag [98]. Furthermore, MVs containing acetylcholinesterase, membrane proteins, glucose membrane transporters, and amino acid transporters might contribute to RBCs membrane remodeling during maturation [99]. The overall balance of physical and chemical changes in stored blood may contribute to immunomodulation and potential adverse effects in patients who have received older blood, and EVs may be key mediators of immune modulation in transfusion recipients [100,101].

Intercellular communications between neighboring and distant cells are crucial for the survival of cells and responding to paracrine and endocrine signaling [102]. Cell-to-cell communication is an essential component in mammalian development and preservation of homeostasis, ensuring efficient responses to alterations or threats within the environment surrounding host cells [103]. The classical modes of intercellular communication involve cell junctions, adhesion contacts, and soluble factors that can act upon the same cell where they are produced, or upon neighboring cells, or may even act over long distances by secreting soluble factors, such as hormones and cytokines to facilitate signal propagation [3,104]. In multicellular organisms, cells can communicate via extracellular molecules such as nucleotides, lipids, or proteins, and cells can release various types of membrane vesicles into the extracellular space that differ in origin, size, morphology, and content to best adapt to their surrounding microenvironment [105,106,107,108]. EVs released from parental cells may interact with target cells, leading to the subsequent influence of target cell behavior and phenotype features [109]. In plasma, we have isolated EVs from adult, children, and animals exposed to sleep apnea [107,110,111,112,113,114,115].

In general, EVs are a heterogeneous population of vesicular bodies of cellular origin that derive either from the endosomal compartment (exosomes, 30–150 nm) or as a result of shedding from the plasma membrane (microvesicles 100–1000 nm, large oncosomes (1000 × 10,000 nm) and apoptotic bodies (100–5000 nm) [3]. Furthermore, EVs can be secreted from all tissues and organs in both health and disease conditions, and these include exosomes, epididymosomes, prostasomes, ectosomes, apoptotic bodies, microvesicles, and more recently oncosomes [96,97,108], and analysis of the EV contents can provide information about differentiation/functional state of parental cells [116]. These EVs have been found in all biological fluids including plasma, urine, saliva, semen, bronchoalveolar lavage, bile, ascitic fluid, breast milk, cerebrospinal fluid, and RBCs [117,118,119]. The first observation of EVs was in platelet-free serum in 1946 [120]. EVs carry a variety of cargo, including RNAs, proteins, lipids, and DNA, which can be taken up by other cells, both in the direct vicinity of the source cell and at distant sites in the body via biofluids, and elicit a variety of phenotypic responses [114,119,121,122,123].

Numerous methods have been used for EVs isolation from the body fluids and cell cultures based on their shape, density, size, or surface components; however, EVs isolation from RBCs are very limited. There is no gold standard for EVs or exosomes isolation from all cell sources or even from one sample type. Recently, the EVs research established a guideline for these processes [122,124]. The most common methods for isolation of EVs or exosomes can be applied from one source to another with some modification. The following are the most common methods used for isolating EVs: ultracentrifugation, sequential centrifugation, density gradient, filtration, sucrose gradient precipitation, polyethylene glycol, acetate, size-exclusion column (Sepharose gel, Sephadex affinity-based capture), and magnetic beads [110,111,112,113,125,126,127,128]. Each of these methods has advantages and disadvantages which include time-consuming, labor-intensive, and expensive equipment.

These methods can be applied to isolate EVs or exosomes from RBCs. The most suitable methods are size-exclusion column or a combination of different methods because the red color in RBCs can cause issues for isolation Figure 3. EVs were isolated from mice RBC-derived conditioned media by differential centrifugation and ultracentrifugation [129,130]; however, ultracentrifugation and precipitation methods are not recommended due to the high red color in the isolation. Ultracentrifugation is labor intensive and requires large sample volumes and produces a relatively low yield of enriched EVs [131]. Numerous alternative isolation methods have been developed including density gradients (DG) and size exclusion chromatography (SEC). Although DG and SEC usually result in high-purity EV, these protocols are time consuming, characterized by poor yields and suitable for small input volumes [124,132]. Immunoaffinity-based approaches can also be used for EVs isolation [133]. Most of these devices utilize immunoaffinity-based isolation methods using magnetic beads or microfluidic chips coated with EV-specific antibodies such as CD9, CD63, and CD81, which isolate EVs with relatively high purity. Over the last few years, several guidelines were reported for exosomes isolation and characterization from different cell sources [122,124,134,135].

Many new technologies and devices continue to be developed based on the importance of the field, and there are several challenges that still need to be addressed. These include improvements in yield, purity, reproducibility of methods involving EVs isolation from biological samples, automation of the EV-enrichment process enabling its rapid application to clinical settings, and facile retrieval of bioactive EVs compatible with downstream molecular analysis. Several investigators have reported about the isolation and the characterization of EVs or microparticles from red blood cells [136,137,138,139] and the release mechanism of MVs in RBCs, and their properties in response to ATP depletion, oxidative stress, and storage were also suggested [140,141,142]. More recently, isolation of EVs derived from plasmodium-infected and non-infected RBCs as targeted drug delivery vehicles has been reported [117]. In addition, an increased level of RBCs-derived MVs in the circulating blood of patients with SCA, thalassemia, and glucose-6-phosphate-deficiency (G6PD) have been observed [143,144]. The most studied RBC-derived EVs are formed during RBCs storage [145,146]. The concentration of EVs is affected by component preparation methods, storage solutions, and inter-donor variation [139]. The precise molecular mechanisms and signaling pathways responsible for formation and release of RBCs-derived EVs are unknown. As the direct result of the continuous formation of EVs, aside from the gradual decrease in RBCs size, both stored RBCs as well as RBCs-derived EVs display increasing levels of PS on their surface [96].

The quality and clinical impact of stored and differentially manufactured red cell concentrates from different donor groups is a major concern, and as exosomes are markers of cellular activation or degradation, a recent study investigated the utility of exosomes screening to characterize the effects of RBCs production methods and storage [147]. The most used EV isolation methods include ultracentrifugation, density gradient centrifugation, size exclusion chromatography, and polymer-based precipitation, with each varying in yield of EVs, the depletion of lipoproteins and protein contaminants, labor-intensity, and cost of the procedure. Therefore, the choice of the EV isolation method used should depend on the amount of starting material together with the downstream application and would be influenced by the need to remove all or only distinct groups of non-EV serum components Figure 3.

There are different methodologies to characterize and/or quantify EVs as shown Figure 4. EVs morphology can be characterized using conventional electron microscopy which provides information on their structure and size as a cup-shaped morphology [112,113]. Quantitative methods to detect EVs include nanoparticle tracking analysis (NTA) and tunable resistive pulse sensing (TRPS) [148,149]. NTA and TRPS are able to provide information about the size and concentration of EVs in a sample but are unable to identify EV subtypes.

Exosomes in the circulation carry biomarkers originated from the donor cells, and exosomes contain various markers which indicate their origins, e.g., CD235a (GPA) for RBCs, CD41 for platelets, and CD11c for dendritic cells [150,151].

For example, RBCs CD59 and decay-accelerating factor (DAF) are known as complement inhibitors and signaling of CD47 (CD47 inhibits RBCs phagocytosis via macrophages by binding to the inhibitory receptor signal regulatory protein alpha) (SIRPα) and SHPS-1 molecules on the cell surface to protect themselves against endogenous elimination [152,153]. Thus, the presence of such proteins on the surfaces of RBC-EVs may help to escape from the clearance by macrophages, if they carry CD47 on their surfaces [154,155].

The mechanism of the effects of IH on the research of microvesicle and RBCs is shown is Figure 5. In this Figure, IH leads to activated ROS, which increases releases of Ca^2+^ and subsequently release of EVs into the circulation. The Ca^2+^ influx via nonspecific cation channels promote the activation of calpain protease and scramblase and inhibits flippase leading to PS externalization, cytoskeletal proteolytic degradation, and band 3 aggregation, all of which promote RBC membrane vesiculation [156].

## 6. Extracellular Vesicles Cargo

EVs are formed in the endocytic compartment of parent cells in MVB and, upon MVB fusion with the cellular membrane, are released into tissue spaces and the bloodstream [119]. The composition of exosomes cargo varies depending on their origin, meaning that the cell from which they come is a critical factor in determining the exosomes. It has been reported that the recorded information of ExoCarta (http://microvesicles.org, accessed on 12 September 2016) contains 41,860 protein entries, 9769 mRNA entries, and 1116 lipids entries, while Vesiclepedia (http://microvesicles.org accessed on 12 September 2016) have 349,988 (protein entries), 27,646 (10,520 (miRNA entries), and 639 (lipid entries). EVs are highly enriched in proteins with various functions, such as tetraspanins (CD9, CD63, CD81, and CD82) that are involved in cell penetration, invasion, and fusion events; heat-shock proteins (HSP70, HSP90) are involved in antigen binding and presentation; MVB formation proteins (Alix, TSG101) are involved in exosome release [157]. Among these proteins Alix, Flotillin, and TSG101 can participate in exosome biogenesis rendering exosomes distinct from the ectosomes released upon plasma membrane shedding. The presence of known cellular proteins in exosome preparations from various cellular sources have been analyzed by Western blotting and by fluorescence-activated cell sorting (FACS) analysis of exosome-coated beads [107,112,115] based on their components including integrins and tetraspanins (CD63, CD9, and CD82) Figure 4. The function of these proteins in EVs is unknown at present. Importantly, these studies also showed that exosomes are clearly distinct from the vesicles that are produced by apoptotic cells, and they are only secreted by living cells.

### 6.1. Extracellular Vesicle RNAs

The presence of RNA in EVs was first described in 2006 for murine stem cell derived EVs and in 2007 for murine mast cell-derived exosomes taken up by the human mast cells [158,159]. In EVs, various types of RNA have been identified including messenger RNA (mRNA), transfer RNA (tRNA), small interfering RNA (siRNA), long-non-coding RNA (lncRNA), and miRNA [160]. In addition, EVs contain defined patterns of mRNA, miRNA, long non-coding RNA, and genomic DNA, and therefore, their contributions to cellular trafficking consist of the transfer of genetic information that in turn induces transient or persistent phenotypic changes in the recipient cells [107,113,119,158,161,162,163,164]. Most circulating small RNAs (sRNAs) are contained within lipids or lipoprotein complexes, apoptotic bodies, or EVs that efficiently protect them from degradation by serum ribonucleases.

The discovery of RNA molecules in blood EVs provides protection to RNA molecules from being degraded by RNAses and lead to an increased interest in the profiling of RNAome in blood EVs under different conditions [3,165,166]. EVs also contain different patterns of RNAs that can be incorporated into recipient cells, and data from next-generation sequencing (NGS) showed that these RNAs contain miRNAs which were the most abundant in human plasma-derived exosomal RNA species, making up 42.32% of all raw reads and 76.20% of all mappable reads [167]. Other RNA species including ribosomal RNA (9.16% of all mappable counts), long non-coding RNA (3.36%), piwi-interacting RNA (1.31%), transfer RNA (1.24%), small nuclear RNA (0.18%), and small nucleolar RNA (0.01%). Once miRNAs are packed into EVs or exosomes, they can undergo unidirectional transfer between cells, resulting in the establishment of an intercellular trafficking network, which, in turn, elicits transient or persistent phenotypic changes of recipient cells [3,113,168]. MiRNAs, a class of small non-coding RNAs mediating post-transcriptional gene silencing, are involved in several human diseases, including CVDs and various cancer types. Dysregulation of miRNA functions and levels are associated to numerous human diseases [112,119,169,170].

It has been suggested that human RBCs are a good candidate source for EVs therapies because (i) RBCs lack both nuclear and mitochondrial DNA; (ii) RBCs are the most abundant cell type (84% of all cells) in the body; and (iii) RBCs can be obtained from any human subject readily and have been used safely and routinely for blood transfusions over decades [171,172]. The use of RBC-EVs as a versatile delivery system for therapeutic RNAs, including short RNAs, as well as long RNAs such as Cas9 mRNAs has been demonstrated [173].

### 6.2. Extracellular Vesicle miRNAs

MiRNAs, which contain 19–23 nucleotides non-coding RNAs, are known as a mediator of post-transcriptional regulation, which can negatively regulate the expression of target mRNAs [174]. While the majority of miRNAs are located within the cell, recently, a significant number of miRNAs have been found in the extracellular environment, including various biological fluids and cell culture media, commonly known as circulating miRNAs or extracellular miRNAs [175]. An increasing number of extracellular miRNAs have been detected in exosomes isolated from biological fluids and cell culture media [158]. The EVs cargo specially miRNAs can play an important role in the vesicles for therapeutic targets [111,119,176]. A recent study compared miRNA isolation from RBCs, serum, and exosomes, and they found that 38 miRNAs are identified in serum, exosomes, and RBCs, suggesting that blood may contain EVs derived from RBCs, and serum may contain miRNAs released by RBCs [177]. Another study has detected that 78 miRNAs were found in EVs from stored RBCs, and miR-125b-5p, miR-4454, and miR- 451a were the most abundant miRNAs with potential functions [178]. Furthermore, several studies reported about the isolation of EVs-miRNAs from RBCs including animal and human, and some of these miRNAs are miR-125b-5p, miR-4454, miR-29a-3p, 125b-5p, miR-451, and miR-144 [178,179,180]. Multiple studies highlighted the importance of miRNAs from different organs including bone marrow, lungs, liver, spleen, and kidneys may be potentially used as targets of RBCs miRNAs [181,182].

### 6.3. Extracellular Vesicle Lipids

Lipids are essential components of exosomal membranes, and specific lipids are enriched in exosomes compared to their parent cell. For example, exosomes are enriched in phosphatidylserine (PS), phosphatidic acid, cholesterol, sphingomyelin (SM), arachidonic acid, and other fatty acids, prostaglandins, and leukotrienes, which account for their stability and structural rigidity [183]. Furthermore, exosomes have been proposed to be enriched with sphingolipids involving MVB formation, whereas microvesicles are enriched with phospholipids, including phosphatidylethanolamine and PS corresponding to the lipid compositions of plasma membrane [183,184]. Lipids play several pivotal roles in fundamental biological functions of EVs, including cell membrane structure, chemical signaling, and cholesterol metabolism [185]. Usually, membrane-bounded vesicles derived from mammalian cells are composed of five major classes of lipids, including sphingolipid, fatty acid, glycerolipid, glycerophospholipid, and sterol lipid classes, and each of these lipid classes contains the diversity of lipid species [186]. EVs lipids have been implicated in altering cellular function and signaling including ceramides, sphingomyelins, PS, and phosphatidylcholines [187,188]. For example, ceramides, play an important role in the budding of exosomes, can induce apoptosis in oligodendroglioma cells, and promote macrophage chemotaxis in a mouse model [189,190]. Sphingomyelin is an important component of the plasma membranes and lipid rafts, while phosphatidylglycerols are a known component of pulmonary surfactant [191]. Ceramides, sphingomyelins, and phospholipids are involved in cellular survival and apoptosis, airway remodeling, and immune cell activation [192,193]. Furthermore, the balance between ceramides and sphingomyelins levels has been shown to be critical in regulating cell survival, differentiation, and apoptosis, thus leading to the concept of the sphingolipid rheostat [194,195]. Recent studies reported about RBCs-EVs lipid classes including phospholipid (PL) in blood during storage [196] and blood transfusion [197].

Lipids play important roles in cells and have various biological functions, such as energy storage or serving as precursors for metabolic processes. The dysregulation of lipid metabolism in cancer, neurodegenerative diseases, and CVD has been reported in the literature [198,199]. Lipids have been implicated in multiple aspects of exosome biogenesis and function. Due to the presence of lipid raft-associated proteins, including flotillin-1, in EVs, it is thought that lipid rafts may be influencing selective protein sorting into exosomes [200]. Lipid rafts are rich in cholesterol and glycosphingolipids, acting as platforms for lipid raft-associated protein signaling. Further, the tetraspanins CD9 and CD81 are present in most exosomes preparations, and both have known associations with cholesterol [201], suggesting that their specific incorporation and enrichment in EVs may be due to their associations with lipid rafts. Cholesterols, sphingomyelins, and PS are the major components of lipid rafts, with all three lipids showing an increased abundance in EVs when compared to their secreting cells [183,202].

### 6.4. Extracellular Vesicle Proteins

The initial proteomic analysis indicated that EVs harbor a specific subset of cellular proteins, some of which rely on the cell that secretes them, while others are observed in most exosomes regardless of cell types [203]. The vesicular proteomes can provide diverse information regarding the biogenesis mechanisms and pathophysiological functions of EVs and facilitate biomarker discovery based on the protein signature of the originating cells [204]. Annotation of EVs proteomes according to their subcellular distribution revealed that plasma membrane and cytoplasmic proteins are more commonly sorted into extracellular vesicles compared to those of the nucleus and mitochondrion [204]. Several mechanisms of protein sorting into EVs during their biogenesis have been proposed, for example, ubiquitinylated membrane proteins sorted by endosomal-sorting complexes required for transport; co-sorting by protein interaction; co-sorting by lipid interaction; and nonspecific engulfment of cytosolic proteins [205]. However, the molecular mechanisms by which proteins are loaded into EVs are not fully understood. The co-sorting of cytoplasmic proteins with vesicular cargo proteins via protein–protein interactions, suggesting that direct interactions between cellular proteins are critical for cargo protein sorting during vesicle formation [206].

EVs proteins include (i) membrane transport and fusion related proteins like annexin, Rab-GTPase (Ras-related protein GTPase Rab), and heat-shock proteins (HSPs) including Hsp60, Hsp70, and Hsp90; (ii) tetraspanins (also termed four-transmembrane crosslinked proteins), including CD9, CD63, CD81, CD82, CD106, Tspan8, ICAM (intercellular adhesion molecule)-1; (iii) MVBs related proteins, for instance, ALIX, and TSG101 (the stereotypical biomarker for exosomes characterization); (iv) other proteins, such as integrins (cell adhesion-related proteins), actin, and myosin (participating in cytoskeletal construction) [207,208]. Tetraspanins can facilitate the entry of specific cargos into exosomes, and CD9 mediates the metalloproteinase CD10 loading into exosome [209]. As members of MVBs-related proteins, ALIX and TSG101 are known components of ESCAT machinery and classify the cargo proteins of intraluminal vesicles (ILVs) by recognizing the ubiquitinylated proteins and then arranging them on the plasma membrane as the components of exosomes [210]. HSPs facilitate protein folding and balance of proteostasis and proteolysis acting as the molecular chaperones and play anti-apoptotic roles in tumors [211]. Among the HSPs, Hsp90 is the major intercellular chaperone that ensures proper protein folding and function by interacting with a variety of intracellular proteins. Tumor cells are constantly in a state of stress like hypoxia, acidosis, metabolic, and nutrient deficiency, which promotes the high expression of Hsp90 in various cancer cells. Hsp90 plays a crucial role in promoting tumor growth and metastasis of breast cancer, pancreatic cancer, leukemia, and closely associated with poor prognosis of tumors [212]. Additionally, for exosome-related functions, a recently research showed that membrane deformability of Hsp90 mediates fusion of MVBs and plasma membrane [213]. The EVs lack of extracellular Hsp90α, a key subtype of Hsp90, loses the capacity to carry out the important intercellular communication from tumor cells to stromal cells, promoting cellular motility [214].

EVs are highly enriched in tetraspanins, and these proteins are organized as membrane microdomains termed tetraspanin-enriched microdomains by forming clusters and interacting with a large variety of transmembrane and cytosolic signaling proteins [215,216,217]. Among tetraspanins, CD9, CD63, CD81, and CD82 have a broad tissue distribution, while others are restricted to particular tissues, such as CD37 and CD53 in hematopoietic cells [218]. Immunoelectron microscopy studies have showed that tetraspanins are abundant on various types of endocytic membranes and have been widely used as exosomal markers [219]. Several studies emphasized the importance of tetraspanin CD81 in the assembly of a functional immunological synapse and present in the central supramolecular activation complex (c-SMAC) [220,221]. In addition, CD81 is associated with CD4 and CD8 playing a role in co-stimulatory signals, while mice deficient in CD81 present a delayed humoral response with impaired T- and B-cell activation [222,223]. CD63 is a tetraspanin molecule and an important scaffold protein known to play a role in structuring the immunological synapse, which is critical for effective immune signaling [224].

OMICS studies including lipidomic, genomics, and proteomics are crucial for understanding cellular physiology and pathology; consequently, lipid biology has become a major research target of the post-genomic revolution and systems biology [106] as illustrated in Figure 6. Since, RBC-EVs cargos are connected, it would be much better to isolate these cargoes at the same time finding networks that may play a role in their biological function. The development of omics technologies and increased generation of high-throughput omics data have inevitably called for powerful systematic means to analyze them in an integrated manner. Omics, a new way for exploring the pathogenesis of human diseases, is categorized into genomics, epigenomics, transcriptomics, and proteomics, and multiple authors have reported about RBCs and OMICs integrations [225,226,227,228].

## 7. Extracellular Vesicles Internalization

EVs uptake is a constitutive homeostatic process among many cell types except mature RBCs, which lack the endocytic machinery [229]. The biological function for exosomes or EVs seems to be similar from different sources; however, different functions have been attributed to exosomes depending on their origin, contents, type, and the physiological state [230]. Given the emerging role of EVs in both physiological and pathological conditions, EVs can be uptaken by cells and can stably transfer drugs, such as therapeutic miRNAs and proteins [108,113,114,123,231]. The success of EVs biological purposes is highly dependent on effective delivery of cargoes, which can be accomplished via receptor–ligand interactions, direct fusion of membranes, or internalization by endocytosis [232]. After being internalized, EVs can fuse with the limiting membrane of endosomes, resulting in the horizontal genetic transfer of their content to the cytoplasm of target cells. EVs can be internalized into target cells via the following mechanisms: (a) direct fusion of exosome lipid bilayer with cell plasma membrane, delivering lumen cargo in the cytosol; (b) binding of exosome membrane proteins with cellular receptor inducing intracellular signaling cascades; (c) phagocytosis of exosomes; (d) clathrin-mediated endocytosis; (e) lipid-raft mediated endocytosis dependent on specific ligand-receptor interaction; and (f) caveolin-mediated endocytosis [119,123]. As a result, parental cells can communicate with specific proximal or distal target cells through exosome amplification. Recent studies have shown a wide range of multiple functions of these vesicles in several biological processes including antigen presentation, neuronal communication, blood coagulation, wound healing, and senescent [107,110,233,234,235]. EVs are also playing an important role in pathogenic processes including cancer, autoimmune diseases, inflammation, infection, and metabolic and cardiovascular diseases [108,236,237]. The origin of EVs determines their structure, components, functions, and potential, making these vesicles as diverse as the types of cells that secrete them [238].

EVs can be tracked in vitro and in vivo by labeling them directly after isolation or indirectly by transfecting the EVs-secreting cells with vectors containing imaging reporter genes. Exosome uptake has been monitored mainly using both flow cytometry and confocal microscopy. This direct labelling allowed for the analysis of the dynamic localization of EVs through the labeling with fluorescent lipid membrane dyes. Examples of such dyes include PKH67, PKH26, rhodamine B, DiI, and DiD [239,240,241,242], respectively. The use of GFP-tagged exosomal proteins also (e.g., GFP-CD63) allowed for direct vesicle visualization, confirming their rapid incorporation into recipient cells [243]. The treatment of target cells with either acidic buffers or trypsin allowed for the discriminating between internalized and surface-bound fluorescent vesicles. [243,244]. Indirect labeling using genetically modified cells such as fluorescent proteins (GFP, RFP, or dTomato) or luciferase enzyme-substrates (Gluc, GlucB or Rluc) can be detected [245,246,247]. More recently, nuclear imaging-based techniques such as positron emission tomography (PET), and single-photon emission tomography (SPECT) was introduced for in vitro tracking of EVs [248]. The precise molecular mechanisms and signaling pathways responsible for formation and release of RBCs-derived EVs are unknown. Erythrocytes, platelets, and other cell types are release both extracellular vesicles including exosomes which may have either detrimental or protective characteristics in the setting of IR, and these effects may be mediated by the transfer of miRNA to cardiomyocytes or through ligand–receptor signaling [249].

## 8. Extracellular Vesicles Clinical Application and Therapeutic Potential

Multiple efforts have focused on the development of diagnostic and therapeutic applications that are based on EVs cargo, primarily RNA encapsulated in exosomes or carried in other carrier subtypes [250]. The growing interest in EVs including exosomes and microvesicles as therapeutic applications, particularly in stem-cell-related approaches, has emphasized the need for standardization and coordination of development efforts. EVs are released by various types of cells, and they are relatively stable in the blood circulation and biological fluids [251]. These EVs are promising nanocarriers for clinical use; however, the clinical applications of EVs are still at the early stage, and further investigations are required. Most studies have focused on EVs cargo as biomarkers; nevertheless, none of these biomarkers have been approved by the Food and Drug Administration [252,253].

The advantages of using EVs over synthetic delivery systems including (i) greater stability in the blood because of their natural surface composition [117,254]; (ii) possibly better protection of the encapsulated cargo due to a proteo-lipid architecture [255]; (iii) endogenous cell and tissue targeting features afforded by their adhesion molecules and surface ligands [254]; and (iv) higher biocompatibility allowing improved permeability through biological barriers, including the blood–brain barrier [173]. It has been suggested that EVs may serve as novel tools for various therapeutic approaches, including (a) anti-tumor therapy; (b) pathogen vaccination; (c) immune-modulatory and regenerative therapies; and (d) drug delivery [236,256]. Liposomes have been used for the delivery of therapeutic agents, and exosomes have been proposed as an alternative for the delivery of therapeutic agents because they are composed of natural non-synthetic components, and their size is smaller. Their lipid bilayer structure protects the cargo from degradation, facilitating delivery to its target, and flexibility enables them to cross major biological membranes [257,258]. The complexity, heterogeneity, and functions of EVs under physiological and pathological conditions make them interesting candidates for application in therapies for a wide range of diseases. The use of EVs as vehicles for drug delivery has demonstrated that we can harness their intrinsic advantageous properties to improve drug therapies. Exosomes are found in circulating blood and these vesicles have a half-life of 5–10 min following that period will be removed by tissue-resident macrophages from liver spleen lungs [185,259].

Compared to most other current methods for programmable RNA drug therapies, which are unsuitable for clinical use because of the low uptake efficiency and high cytotoxicity, RBC-EVs show promising prospects [173]. Several factors are considered in order to establish clinical application of EV-based therapeutics including (i) source of the starting material (donor inclusion/exclusion and donor release criteria); (ii) EV-source characterization (donors, donor cells/tissues/fluids, and culture reagents); (iii) EV isolation and storage, including isolation techniques and standardization, purity and impurities, scalability of technology and storage conditions, and quality control; and (iv) molecular and physical characterization (quantitative analyses, counts, and size). EVs have multiple advantages for the delivery of therapeutic cargos because they exhibit specificity, safety, stability, low immunogenicity, and possibility of transportation of substances over their neighbor for long distances [260]. However, the mechanism of EV interaction with cells depends on the recipient cell type, which influences their biodistribution and therapeutic potential [261,262].

RBCs-EVs can be used as potential delivery vehicles for clinical applications due to their autologous property to human bone marrow mesenchymal stem cells [263]. It has been indicated that RBCs-EVs have several features that make them better suited for clinical applications than EVs from other cell types: (i) Blood units are easily accessible. (ii) RBC-EVs are safer compared to EVs from other cell types because the enucleated RBCs contain no DNA, unlike EVs from nucleated cell types which represent a potential risk for horizontal gene transfer. As plasma EVs are heterogeneous with unpredictable contents, RBC-EVs are safer than plasma EVs for allogeneic treatments of cancer because cancer cells and immune cells are known to release large amounts of cancer-promoting EVs into their environment [173,264,265]. (iii) RBCs-EVs are nontoxic; hence, they are nontoxic as compared to the toxic synthetic transfection reagents, for which are typically used. (iv) RBCs-EVs are presumably non-immunogenic via blood type matching, unlike adenoviruses, adeno-associated viruses, lentiviruses, nanoparticles, and various synthetic transfection reagents. (v) RBCs-EVs can be frozen and thawed many cycles without affecting their integrity or efficacy. RBCs-EVs can be developed into stable pharmaceutical products in the future, but further research is urgently needed.

Exploring different cell sources for therapeutic EVs are of interest because the lipid and surface protein composition of exosomes may be crucial to their function, and preservation of these characteristics is very important [266]. EVs isolated from RBCs can be used for NGS to identify biomarkers either miRNAs, mRNAs, or protein as shown in Figure 7. For example, both mRNA and miRNA contained exosomes which are taken up by specific target cells to facilitate disease spreading and pathogenesis [112]. As such, exosomes can provide protection of RNA content from degradation in the environment, enabling a stable source for reliable detection of RNA biomarkers [119,123]. EVs have emerged as an important drug delivery system to deliver therapeutic agents, including proteins and gene therapy for the treatment of many diseases such as CVD [267,268,269]. For clinical applications, future efforts will be focused on the manufacturing process, characterization of EVs, and EVs-associated safety issues. Figure 7 shows potential methods to produce RBCs-EVs, via cell stimulation or in a mechanical way passing through a syringe, and the main mechanisms to introduce the cargo, using the biogenesis process or uploading the drugs with electroporation.

The development of new nanoparticle-based vehicles has become very important, as such systems can improve the therapeutic efficacy of certain drugs by allowing controlled access and administration to target cells. However, the following questions remain unanswered: What are the biological network implications of RBC-EVs cargo? Do the RBC-EVs released by a cell change over time? What additional variables can be considered to define RBC-EVs homogeneity and heterogeneity? What are the contributions of biogenesis pathway and composition to the RBC-EV biology? What are the mechanisms underlying the efficient unpacking of RBC-EVs in target cells? How do RBC-EVs alter cell physiology? Are exosomes or EVs biogenesis in plasma the same as in the RBCs? Are the clinical applications for exosomes from plasma or serum the same as in the RBCs? Why are there EVs or exosomes in the RBCs, and what are their functions? Does RBC-EVs affect the cells by merely binding to the plasma membrane? How do RBC-EVs alter cell physiology? Does RBC-EVs play a role in disease pathogenesis, or are they merely a marker of disease? How RBC-EVs are produced and secreted to the extracellular environment? What is their specific composition? Which physiological and pathological roles do they have? What is the contribution of biogenesis pathway and composition to the definition of RBC-EV biology? Answering some of these questions will advance our knowledge about the mechanisms of RBC-EVs formation and their pathophysiological relevance and may help to shed light on circulating EVs and to translate their application to clinical practice such as intracellular signaling.

In summary, OSA is the most common non-communicable chronic diseases, including CVD, which is associated with endothelial dysfunction. The benefit of CPAP therapy on cardiovascular outcomes remains uncertain. EVs are important players of exchanges between cells, through the communication of various proteins, lipids, and genetic information to alter the phenotype and function of recipient cells. RBCs-EVs need to determine if they play an important role as EVs from plasma. As EVs from plasma, RBCs-EVs are associated with specific cargo from their parental cell that can include RNAs, free fatty acids, surface receptors, and proteins. These cargos can be measured and analyzed for the development of blood-based biomarkers in diverse disease types such as CVD or metabolic disorders. Although there have been many studies trying to clarify the molecular mechanism of how EVs are produced, more efforts are still needed to further elucidate these problems to make the diagnostic and therapeutic potential of EVs a clinical reality. A recent study has focused on the contribution of RBCs-EVs to endothelial function in OSA or IH-related to endothelial dysfunction. Despite the advances in EVs research, there are still many challenges ahead. For example, achieving large-scale production of EVs for clinical use remains the main challenge. RBC-EVs need further research to establish them as a new source and promising approach for practical therapeutics in clinical use. The potential impact on precision medicine initiatives warrants a systematic undertaking to develop new methods to interrogate EVs populations across biological fluids and understanding their therapeutic potential. A greater understanding of the specific molecules on the surface of RBCs-EVs that promote interactions in a tissue-specific way is critical for the potential use of exosomes as a future drug-delivery vehicle.

## Figures and Tables

**Figure 1 ijms-22-04301-f001:**
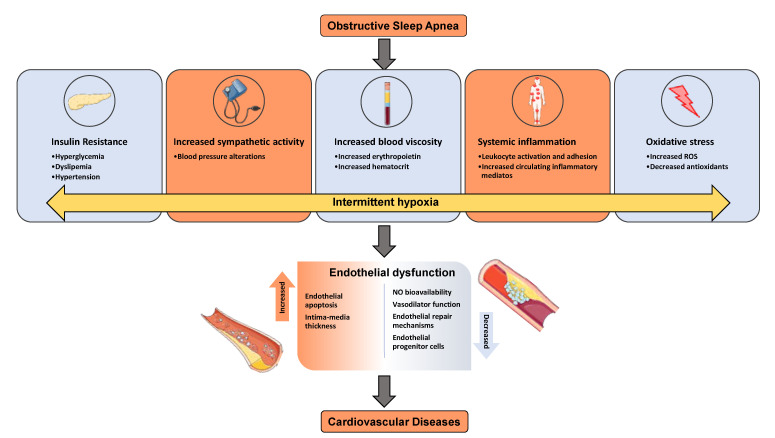
Diagram illustrates the effect of OSA and IH on endothelial dysfunction and cardiovascular diseases and associated mechanisms.

**Figure 2 ijms-22-04301-f002:**
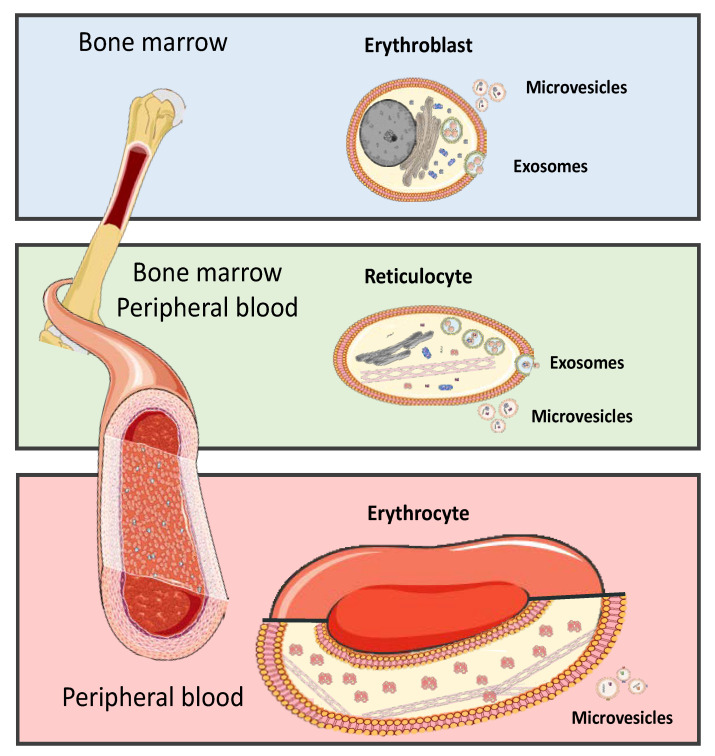
Source of red blood cells. Most blood cells develop from bone marrow cells—the spongy material in the center of the bones—and transfer into the circulation and further release exosomes and microvesicles.

**Figure 3 ijms-22-04301-f003:**
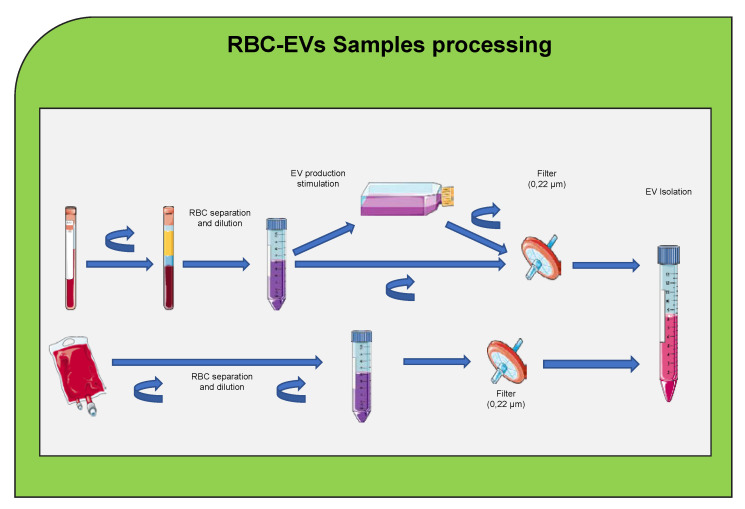
Schema illustrating several methods used to isolate extracellular vesicles (EVs) from red blood cells (RBCs). These methods include ultracentrifugation, size exclusion chromatography, precipitation, and affinity-based isolation.

**Figure 4 ijms-22-04301-f004:**
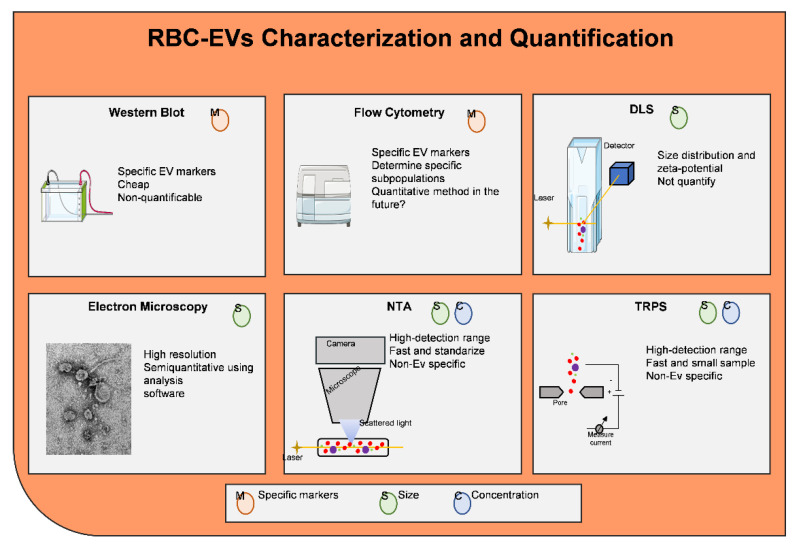
Characterization and quantification of EVs. RBC-EVs can be characterization by western blots, flow cytometry, and electron microscopy. Transmission electron microscopy (TEM), scanning EM (SEM), and cryogenic TEM are largely applied to determine EV features (diameters and morphology). RBC-EV size distribution and polydispersity in a sample can be also analyzed by dynamic light scattering (DLS), which detects the diffusion coefficient of the scattering EVs. Nanoparticle tracking analysis (NTA) allows for the detection of both EV dimensions and concentrations, through the analysis of the EV Brownian motion and the measurement of scattered light (Sc-NTA) or emitted fluorescence (Fl-NTA). Tunable resistive pulse sensing (TRPS) is only measuring particle concentration. Spectradyne’s nCS1Tcan delivers accurate measurements of exosome/vesicle size and concentration in minutes, using microfluidic resistive pulse sensing (MRPS).

**Figure 5 ijms-22-04301-f005:**
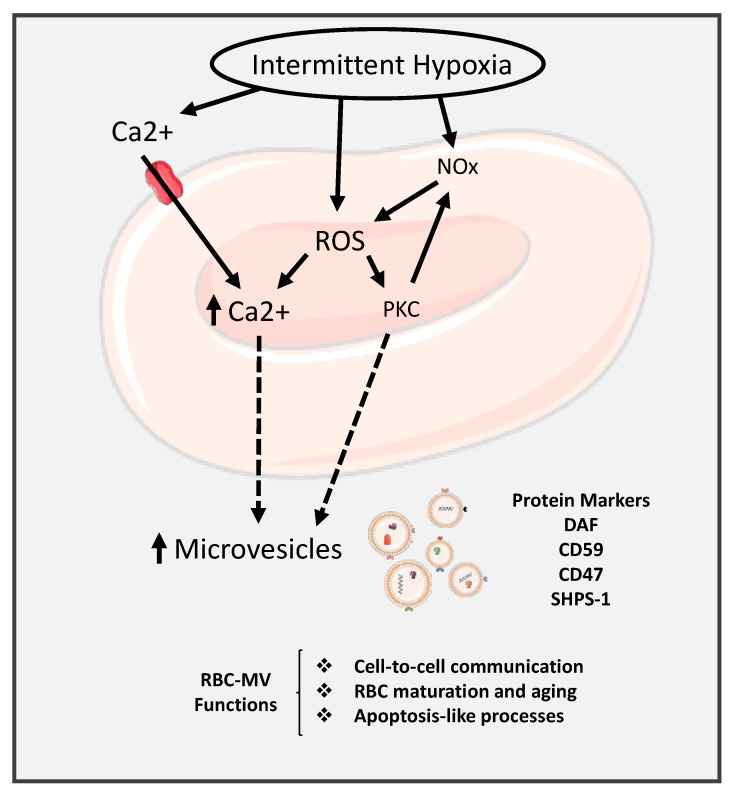
Diagram illustrating the effect of intermittent hypoxia on the release of microvesicles into the circulation. Arrow defined as increase or release.

**Figure 6 ijms-22-04301-f006:**
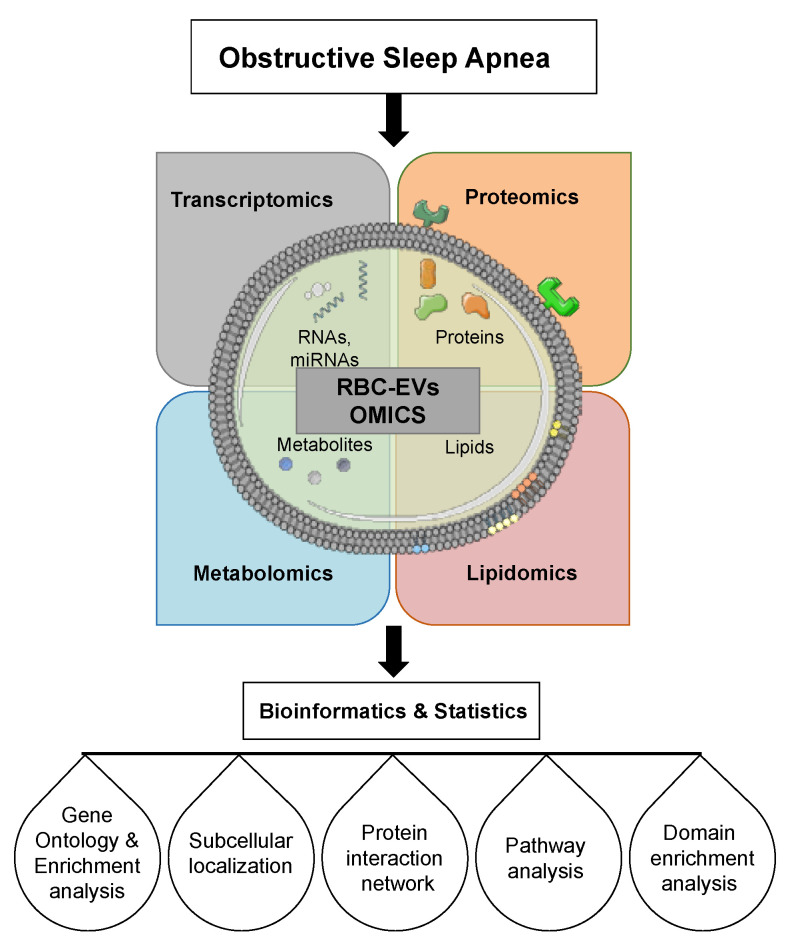
Schema showing RBC-EVs OMICS of OSA. Quality control techniques (NTA, TEM, and imaging flow cytometry) can be performed on the isolated EVs prior to next-generation sequencing of small RNAs and mass spectrometry of proteins. Bioinformatics can be used to identify gene ontology, protein interactions, and pathways analysis.

**Figure 7 ijms-22-04301-f007:**
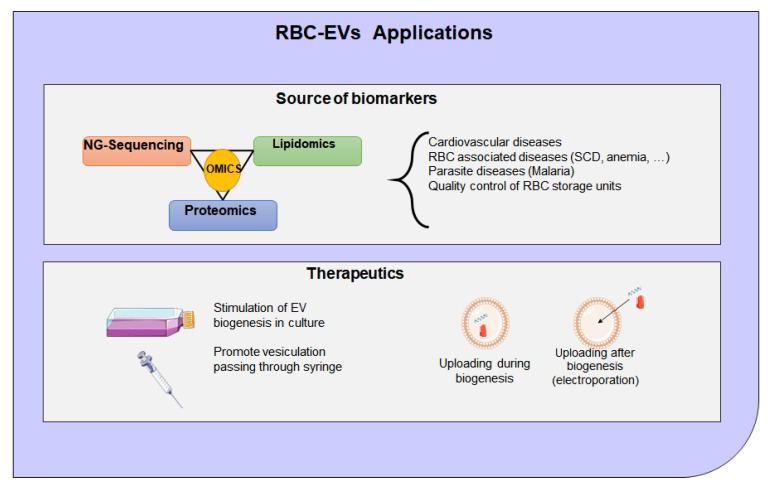
Schema represents the RBC-EVs applications. The biomarkers must be identified first using high-throughput technologies such as next generation sequencing or mass spectrometry. EVs can be collected from cells cultured in vitro, and nucleic acids may be incorporated by electroporation through incubation with the EVs. For therapeutics, cells are treated with a drug, or nucleic acids or proteins and introduced by chemical transfection.

## Data Availability

Not applicable.

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
