# Peer review of "The Mystery of Red Blood Cells Extracellular Vesicles in Sleep Apnea with Metabolic Dysfunction"

_ijms, 2021, doi:10.3390/ijms22094301_

Round 1
Reviewer 1 Report
The manuscript entitled „The mystery of Red Blood Cells Extracellular Vesicles in Sleep Apnea with Metabolic Dysfunction” reviews available literature of red blood cells and their extracellular vesicles on metabolic dysfunction in OSA patients.
I congratulate the authors on this in-depth review. They in great detail describe the undertaken topic. It provides the reader with an easy-to-follow analysis red blood cells extracellular vesicles in sleep apnea with metabolic dysfunction.
However, throughout the manuscript I think the discussion of hypoxia-inducible factor 1 (HIF-1) is missing considering its great effect on the metabolism of OSA patients and erythrocytes themselves. It is highly involved in mediating development and progression of DM2 though hypoxia (doi: 10.3389/fphys.2020.01035). While authors describe the comorbidities, especially effects of IH on blood cells HIF-1 is also highly involved (doi: 10.3389/fneur.2018.00635). Furthermore, while discussing reaction of erythrocytes to hypoxia and their further effects it is important to mention that HIF-1 causes upregulation of erythropoietin (EPO) and is a critical factor responsible for increased production of red blood cells. Therefore, it should not be overlooked how HIF-1 is increased in OSA patients (doi: 10.5664/jcsm.8682, doi: 10.17305/bjbms.2016.1579, doi: 10.20452/pamw.15104, doi: 10.3390/jcm9051599, doi: 10.1111/jsr.12995).
Author Response
Dear Reviewer,
We thank you for your comments and suggestions about HIF. We integrated our responses in the manuscript, and they're highlighted in Red color.
Best regards
Khalyfa
Furthermore, another proposed molecular mechanism is hypoxia-inducible factor 1 (HIF-1α) which is based on the oxygen-sensitive α-subunit of HIF-1α -a key regulator of oxygen metabolism which plays an important role in OSA and metabolic dysfunction [44]. It has been indicated that HIF-1 is highly involved in mediating the development and progression of type 2 diabetes ((T2DM) through hypoxia [44]. In addition, HIF-1 is involved in the effects of IH on blood cells [45]. Several studies in OSA animal and cell models have been shown an increased expression of HIF-1a [46, 47]. Other investigations showed that OSA patients increased HIF-1a levels in comparison to healthy control [48-52].
Thus, HIF-1 may cause upregulation of erythropoietin (EPO) and is a critical factor responsible for increased production of red blood cells [84].
Reviewer 2 Report
The manuscript entitled “The Mystery of Red Blood Cells Extracellular Vesicles in Sleep Apnea with Metabolic Dysfunction” is an interesting literature review integrating pathological features from obstructive sleep apnea syndrome and cardiovascular comorbidities with the metabolic consequences of endothelial dysfunction to the molecular aspects with extracellular vesicles of RBCs. Being not particularly familiar with the more molecular aspects, I find this paper interesting bridging molecular aspects and clinical applications. It has a high pedagogic value describing this issue in a comprehensive way for a non-specialist and with the use of accessible illustrations that are complementary. The organization of this paper into 14 sections seems a non-optimal structure and it could benefit to organize some of the chapters into regrouping subheadings (the more pathological clinical aspects, metabolic consequences, RBS, EV and perspectives).
Author Response
Dear Reviewer,
We thank you for your suggestions about combined some sections together due to the length of the mansucrpt. We changed as suggested.
Best Regards
Khalyfa